# Ultralong UV/mechano-excited room temperature phosphorescence from purely organic cluster excitons

Xuepeng Zhang[1], Lili Du[3], Weijun Zhao[1], Zheng Zhao[1], Yu Xiong[1], Xuewen He[1], Peng Fei Gao[1], Parvej Alam[1], Can Wang[4], Zhen Li[4], Jing Leng[5], Junxue Liu[5], Chuanyao Zhou[5], Jacky W.Y. Lam[1], David Lee Phillips[3]*, Guoqing Zhang[2]* & Ben Zhong Tang[1,6]*

Purely organic room temperature phosphorescence (RTP) has attracted wide attention recently due to its various application potentials. However, ultralong RTP (URTP) with high efficiency is still rarely achieved. Herein, by dissolving 1,8-naphthalic anhydride in certain organic solid hosts, URTP with a lifetime of over 600 ms and overall quantum yield of over 20% is realized. Meanwhile, the URTP can also be achieved by mechanical excitation when the host is mechanoluminescent. Femtosecond transient absorption studies reveal that intersystem crossing of the host is accelerated substantially in the presence of a trace amount of 1,8-naphthalic anhydride. Accordingly, we propose that a cluster exciton spanning the host and guest forms as a transient state before the guest acts as an energy trap for the RTP state. The cluster exciton model proposed here is expected to help expand the varieties of purely organic URTP materials based on an advanced understanding of guest/host combinations.

[1] Department of Chemistry, Hong Kong Branch of Chinese National Engineering Research Centre for Tissue Restoration and Reconstruction, Division of Life Science and State Key Laboratory of Molecular Neuroscience, Institute for Advanced Study, Institute of Molecular Functional Materials, The Hong Kong University of Science and Technology, Clear Water Bay, Kowloon, Hong Kong, China. [2] Hefei National Laboratory for Physical Sciences at the Microscale, University of Science and Technology of China, Hefei 230026, China. [3] Department of Chemistry, The University of Hong Kong, Pokfulam Road, Hong Kong, China. [4] Department of Chemistry, Wuhan University, Wuhan 430072, China. [5] State Key Laboratory for Molecular Reaction Dynamics, Dalian Institute of Chemical Physics, Chinese Academy of Sciences, Dalian 116023, China. [6] Center for Aggregation-Induced Emission, SCUT-HKUST Joint Research Institute, State Key Laboratory of Luminescent Materials and Devices, South China University of Technology, Guangzhou 510640, China. *email: phillips@hku.hk; gzhang@ustc.edu.cn; tangbenz@ust.hk

Featuring the involvement of triplet excited state[1,2], long emission lifetime and large Stokes shift, organic phosphorescence shows great application potentials in a variety of fields such as high-efficiency OLEDs, data-encryption technologies and background-free imaging[3–5]. The quenching propensity of triplet excitons at room temperature, however, tends to make organic phosphorescence only available under cryogenic temperature[1,2], which inevitably hinders practical applications. To achieve more useful room temperature phosphorescence (RTP), a routine method is to incorporate noble metals such as Ir, Pt, and Ru to form organometallic complexes, which yet face issues such as cost, toxicity, and instability against moisture[4]. In comparison, purely organic RTP materials without metals are promising in tackling such problems and thus have attracted particular attentions over the past decade[4–23]. Promoting the intrinsically spin-forbidden intersystem crossing (ISC) is fundamental to realize efficient purely organic RTP. The most common strategy is to introduce functional groups that can facilitate spin-orbit coupling (SOC), such as carbonyl, heteroatoms (O, N, P, etc.) and halogens (Br and I). However, these moieties tend to increase rates of $S_n$–$T_n$ and $T_n$–$S_0$ electronic transitions simultaneously. As such, an inherent dilemma exists that increasing RTP efficiency always leads to decreased RTP lifetime. So far, efficient purely organic RTP with a lifetime longer than 500 ms is still rare[6,23]. This situation may not meet the demand for RTP with both high efficiency and ultralong lifetime in many application fields, such as anti-counterfeiting, novel digital display and time-resolved bioimaging.

To pursue efficient purely organic RTP with ultralong lifetimes, one may be inspired by nature. Nature is abundant in inorganic materials such as luminous pearls and glow-in the dark stones whose noticeable afterglow can last for hours or even longer. Though the precise mechanism of the afterglow in these materials remains ambiguous, it is generally accepted that impurity is responsible and three steps may be involved: (1) excitation of the guest species (impurity), (2) trapping of the excited electrons by defects in host lattices, (3) slow charge recombination of the trapped electrons by thermal energy[24]. Mimicking this guest/host architecture of minerals, purely organic afterglow lasting for more than one hour has been realized recently via doping one electron-donating guest in an electron-withdrawing host[25]. However, the ultralong organic luminescence based on charge separated state may lack versatility. Therefore, we tend to realize and expand the scope of efficient ultralong luminescence based on a more precise and versatile mechanism of molecular phosphorescence, i.e., URTP emitted from triplet excited state, in organic guest/host system.

Purely organic RTP based on guest/host architecture has been reported using co-crystallization or other methods[26–31], whereas almost all reported cases still rely on conventional wisdom of external or internal heavy atom effect from being in close proximity to Br or I[32]. On the one hand, these species are usually unstable due to photo-homolysis propensity of C–Br and C–I bonds[33,34]. On the other hand, they tend to have lifetimes <100 ms due to accelerated $T_n$–$S_0$ transitions. Most importantly, it is generally considered that the guest undergoes the excitation and emission processes, while the host only plays passive roles such as diluting and rigidifying the guest to suppress the non-radiative decay. So far, no investigation has been carried out on detailed photophysics, especially electronic transition processes concerning the host species. Inspired by inorganic afterglow materials, where both host and guest play active roles in electronic transitions, we reason that certain organic counterparts must exhibit similar traits in the solid-state guest/host systems. However, suitable characterization methods must be implemented to study how organic host and guest interact in the excited states.

Therefore, in this study we set out to answer the following questions in organic guest/host systems: (1) Are host and guest molecules acting individually or synergistically in the photoexcited electronic transitions? (2) Are there any transient species formed upon photo-excitation? It remains challenging to find the answers by routine characterizations, such as steady-state emission, excitation, and absorption spectroscopies, which limit the understanding and development for more RTP systems. Herein, in short, we aim to obtain efficient purely organic URTP by guest/host strategy, drawing inspiration from inorganic afterglow materials. Meanwhile, we hope to shed light on RTP photophysics in the solid state by means of ultrafast spectroscopy to show the multiple roles the host molecules play.

To study the model systems, commercially available and scalable molecules are given a high priority. In addition, compounds containing photolabile bond, such as C–Br or C–I are excluded[33,34]. Finally, to avoid quenching from photo-induced electron transfer, all guest and host molecules used in the study are electron-deficient. As a proof of concept (Fig. 1), pentachloropyridine (PCP), phthalic anhydride (PA), and 1,2-dicyanobenzene (DCB) are selected as the host species, respectively, while 1,8-naphthalic anhydride (NA) is selected as the guest due to its more red-shifted absorption. We show that after preparation of the solid-state solution, efficient purely organic RTP with ultralong lifetimes is achieved. Most importantly, fs-TA studies reveal that: (1) host and guest molecules play active roles synergistically in the photo-excited electronic transition processes; (2) cluster excitons spanning both guest and host molecules form as transient species before the localized triplet excited state of the guest acts as an energy trap to emit the URTP. In the sense that host and guest species both play active roles in the photophysics, we are able to show that mechanistically similar RTP, vs. inorganic afterglow, exists in purely organic systems. Furthermore, when host molecules are mechanoluminescence (ML)-active, we show that the tested guest/host systems exhibit an extra feature of ML, which ultimately induces notable URTP with almost identical spectroscopic signatures to the UV-excited URTP. In general, combined advantages including simplicity, stability, low cost, scalability, multiple excitation sources, and excellent performances, along with mechanistic insights revealed by ultrafast spectroscopy of current guest/host URTP system are expected to attract wide attentions from both material and scientific communities.

## Results

**Luminescent properties of single components**. To avoid the possibility of any impurity interference, stringent purification methods such as column chromatography followed by triple recrystallization and HPLC measurement were used for all samples (Supplementary Fig. 1). For comparison, optical properties of the individual components at room temperature and 77 K were studied (Fig. 2 and Supplementary Figs. 2–13), respectively. As shown in Fig. 2, PCP crystals show intense green afterglow at room temperature with a duration of ~1 s ($\lambda_{ex}$ = 254 nm). The steady-state photoluminescence (PL) spectrum shows that PCP crystals exhibit almost pure phosphorescence, i.e., only a very weak fluorescence band is present around 350 nm with a lifetime ($\tau$) of 0.6 ns, while much more intense emission band is centered at 510 nm with a lifetime of 80 ms. The shoulder peaks are apparent at both the fluorescence and phosphorescence bands presumably due to vibrations of aromatic skeletons in PCP molecules. The delayed PL spectrum ($\Delta t$ = 50 ms) filters out of the fluorescence band around 350 nm and is almost identical with the band centered at 510 nm with vibrational peaks in the steady-state spectrum, further verifying the phosphorescence nature of

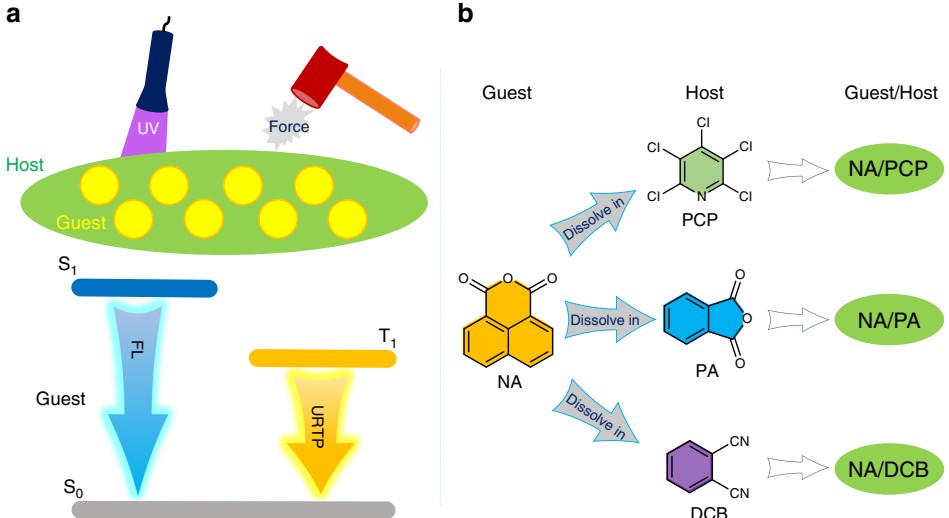

**Fig. 1** Design of the guest/host system. **a** Schematic illustration of the guest/host system, i.e., solid-state solutions where guest is the solute and host is the solvent, showing UV/mechano-excited ultralong room temperature phosphorescence (URTP). FL: fluorescence. **b** Compound structures studied in this work. Abbreviations: PCP for pentachloropyridine, PA for phthalic anhydride, DCB for 1,2-dicyanobenzene and NA for 1,8-naphthalic anhydride

the green afterglow. At 77 K (Supplementary Fig. 6), PCP crystals exhibited blue-shifted phosphorescence (Max $\lambda_{em} = 440$ nm) which is likely due to reduced thermal relaxation from higher energy levels of triplet excited states. Though highly emissive in the solid state at RT with an overall quantum yield ($\phi$) of 36.0%, PCP was almost dark in solution with a $\phi < 1\%$ (Supplementary Fig. 2), indicating its aggregation-induced emission (AIE) property[35]. Interestingly, in addition to strong UV-excited RTP, PCP also shows conspicuous ML[36–43]. When PCP crystals were crushed with mechanical force, persistent green luminescence with a duration of ~1 s was observed without UV excitation. The ML was so strong that it was observable under soft room lighting (Fig. 2a, inserted photo) (Supplementary Movie 1). The ML spectrum of PCP almost superimposes with that of PL except for its less obvious vibrational structures at the fluorescence band around 350 nm. When turned into fine powders after repeated crushing, PCP hardly exhibits ML after being subjected to further mechanical forces. The XRD data of the PCP powder (Supplementary Fig. 23) show extra peaks than do those of the PCP single crystals, indicating that the PCP powders are crystals with much smaller sizes rather than amorphous. Therefore, it is concluded that below a certain limit of crystal size, the PCP solid is no longer mechanoluminescent[36,37].

As for the PA crystals, only extremely faint blue emission was observed under the 254 nm UV irradiation at room temperature. No afterglow could be observed at all after the UV irradiation had ceased. The PL spectrum of the PA crystals exhibits a unimodal peak centered at 342 nm ($\tau = 1.3$ ns, $\phi = 1.0\%$) tailing to the blue region, which is consistent with the very weak blue emission observed by naked eyes. The PA crystals also exhibited weak blue ML upon crushing by mechanical force; yet the emission was so weak that it could only be noted in the dark (Fig. 2b-right). Though the blue emission of the DCB crystals was also weak to naked eyes under UV irradiation at room temperature, the overall quantum yield of the DCB crystals was as high as 19.3%. Steady-state PL spectrum revealed that the major emission peak of the DCB crystals was in the UV region, centering at 335 nm ($\tau = 8.0$ ns), and only very weak tail was present beyond 400 nm. After UV irradiation ceased, very faint yellow–green afterglow lasting for ~3 s was observed. Correspondingly, the delayed PL spectrum was centered at 550 nm with a lifetime of 163 ms. The RTP of the DCB crystals was so weak compared to its fluorescence that the

phosphorescence peak could not be noticed in the steady-state PL spectrum. In addition, DCB crystals exhibited strong blue flash-like ML which could be observed under soft room lighting (Fig. 1c and Supplementary Movie 2). Similar to PCP and PA, the ML spectrum of DCB is also nearly identical to the PL spectrum. On the other hand, NA crystals showed intense blue fluorescence with bimodal peaks at 434 nm ($\tau = 7.3$ ns) and 475 nm ($\tau = 6.7$ ns) respectively at room temperature (Supplementary Fig. 4) with an overall quantum yield of 9.0%, where no RTP or ML was observed.

As for the ML mechanism, the mostly accepted explanation is excitation through electrification[36–43]. In our cases, the ML spectrum of PCP, PA, and DCB all superimposes with their PL spectrum respectively and no peak characteristic of $N_2$ is present. Therefore, it is rational that the crystals themselves instead of $N_2$ are excited and emit the ML[38]. Moreover, PCP ($\phi_{crystal} = 36.0\%$, $\phi_{solution} < 1\%$) is AIE-active and DCB ($\phi_{crystal} = 19.3\%$, $\phi_{solution} = 5.4\%$) shows aggregation-induced emission enhancement (AIEE). Meanwhile, PCP and DCB both exhibit strong ML observable under soft room lighting. In contrast, the PL ($\phi_{crystal} = 1.0\%$) and ML of PA are both very weak. These results indicate that AIE or AIEE property may have a positive effect on the ML intensity[35,39].

**PL properties of the guest/host system**. Under 254 nm UV irradiation, the initial intense green emission for pure PCP crystals vanished in NA/PCP prepared from melt-casting (Supplementary Methods). Instead, as shown in Fig. 3a, an intense off-white emission was observed and a strong yellow afterglow lasting for ~5 s was noticed after the UV irradiation had ceased. The steady-state PL spectrum of NA/PCP shows two main bands centered at 400 nm and 540 nm, respectively. The emission of pure PCP crystals at 350, 478, and 510 nm is still present in the spectrum of NA/PCP, most likely due to the uneven distribution of NA in PCP. Similarly, emission at 440 nm was mainly ascribed to fluorescence of the undissolved NA crystals in the blend. The newly emerged band centered at 400 nm with shoulder peaks at 356, 376 and 418 nm and the more intense one centered at 540 nm with shoulders at 585 and 636 nm were ascribed to species well dispersed at the molecular level, i.e., NA/PCP solid solution where NA is the solute (guest) and PCP is the solvent (host). The lifetimes at 400 and 540 nm were measured to be

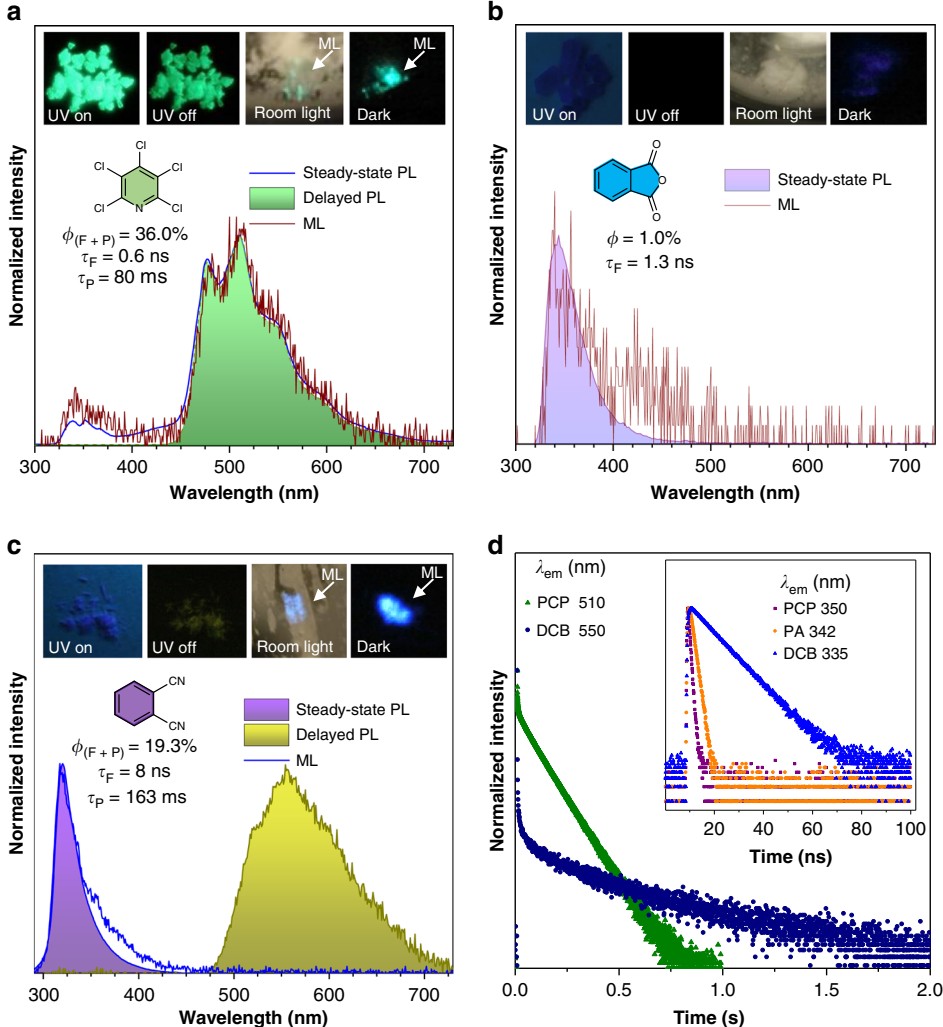

**Fig. 2** Luminescent properties of the host molecules. **a** Steady-state photoluminescence (PL) spectrum, delayed PL spectrum ($\lambda_{ex} = 254$ nm) and mechanoluminescence (ML) spectrum of PCP crystals at ambient conditions. Insert: photos of PCP crystals under 254 nm UV light irradiation and after the UV irradiation ceased under ambient conditions (left); photos of ML produced by crushing PCP crystals on the wall of a glass vial under soft room lighting and in the dark respectively (right). **b** Steady-state PL spectrum ($\lambda_{ex} = 254$ nm) and ML spectrum of PA crystals at ambient conditions. Insert: photos of PA crystals under 254 nm UV light irradiation and after the UV irradiation ceased (left); photos of PA crystals being crushed on the wall of a glass vial under soft room lighting and in the dark respectively (right). **c** Steady-state PL spectrum, delayed PL spectrum ($\lambda_{ex} = 254$ nm) and ML spectrum of DCB crystals at ambient conditions. Insert: photos of DCB crystals under 254 nm UV light irradiation and after the UV irradiation ceased (left); photos of ML produced by crushing DCB crystals on the wall of a glass vial under soft room lighting and in the dark respectively (right). For delayed spectra in (a) and (c), delay time ($\Delta t$) = 50 ms. **d** Time-resolved PL decay curves of PCP crystals at 510 nm and DCB crystals at 550 nm ($\lambda_{ex} = 254$ nm). Insert: time-resolved PL decay curves of PCP crystals at 350 nm, PA crystals at 342 nm and DCB crystals at 335 nm ($\lambda_{ex} = 280$ nm). $\phi_{(F + P)}$: overall quantum yield; $\tau_F$: fluorescence lifetime; $\tau_P$: phosphorescence lifetime

2.2 ns and 363 ms (decay curves are shown in Fig. 3d), indicating their fluorescence and phosphorescence characters respectively. The delayed spectrum at a delay time ($\Delta t$) of 50 ms filters out fluorescence peaks < 450 nm and yet phosphorescence peaks of PCP crystals at 480 and 510 nm are still present. Instead, only ultralong emissions at 540, 585, and 636 nm remain in the delayed PL spectrum at a delay time of 1 s, which is consistent with the ultralong yellow afterglow lasting for ~5 s observed by naked eyes. Given the overwhelming ratio of phosphorescence band in the steady-state PL spectrum and the high overall quantum yield ($\phi = 17.3\%$) of NA/PCP, it is clear that the ultralong yellow RTP of NA/PCP is rather efficient. Therefore, by combining the guest and host molecules in the solid state, purely organic RTP with both high efficiency and ultralong lifetime ($\tau > 100$ ms) can be realized.

As for the NA/PA system (Fig. 3b), similarly persistent yellow RTP with an even longer duration of ~7 s was observed. However, in the steady-state PL spectrum, the phosphorescence ratio is lower than that of NA/PCP. The fluorescence band of NA/PA is centered at 407 nm ($\tau = 6.6$ ns) with shoulders at 356, 376, and 418 nm. While the phosphorescence band, as demonstrated by delayed PL spectrum and lifetime decay curve (Fig. 3d), is centered at 540 nm ($\tau = 492$ ms) with shoulders at 586 nm and 638 nm and the overall quantum yield of NA/PA was measured to be 5.9%. When the host molecule was changed to DCB, the obtained NA/DCB exhibits persistent yellow afterglow lasting for ~9 s after UV ceased. Steady-state PL spectrum shows main fluorescence band centered at 409 nm ($\tau = 3.9$ ns) with shoulders at 370 nm and 392 nm and phosphorescence band centered at 540 nm ($\tau = 603$ ms) with shoulders at 586 and 638 nm. The

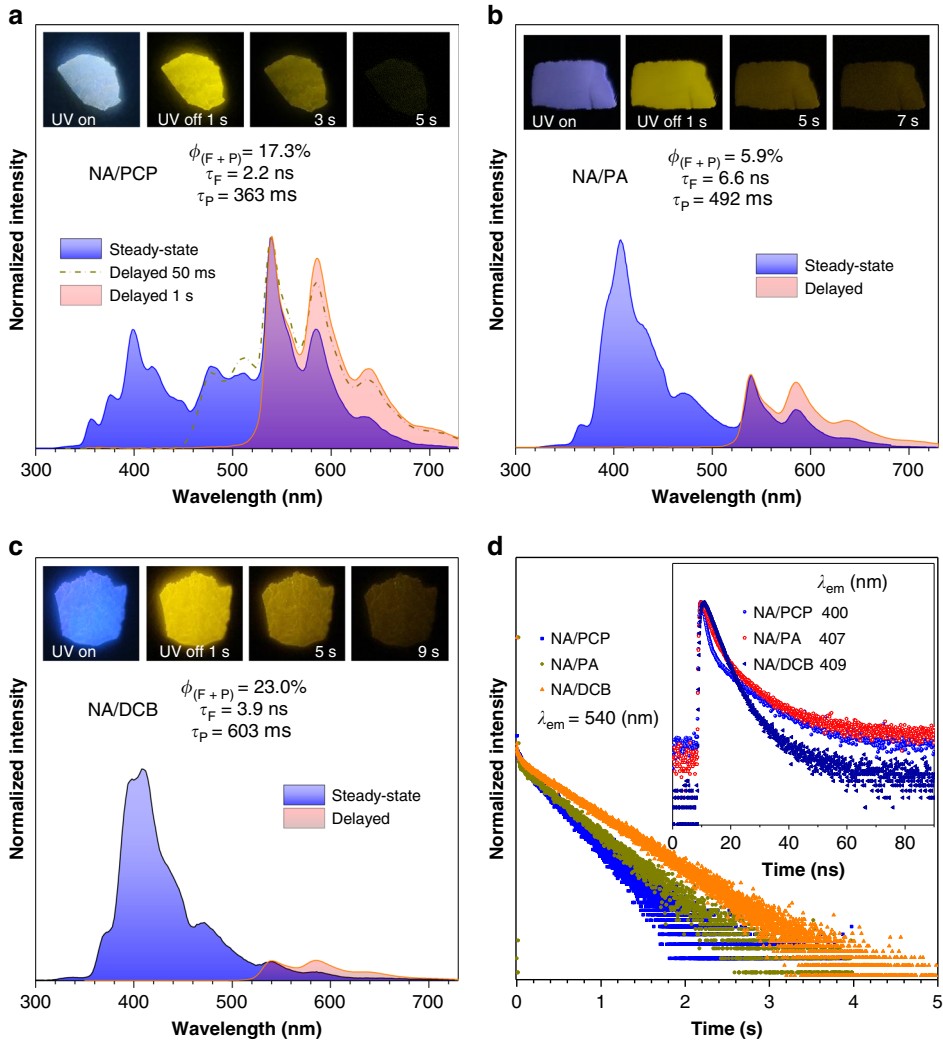

**Fig. 3** Photoluminescence (PL) properties of the guest/host system. **a** Steady-state and delayed PL spectra of NA/PCP ($\lambda_{ex} = 254$ nm) at ambient conditions. Insert: photos of NA/PCP under 254 nm UV light irradiation and after UV irradiation ceased for different times. **b**, **c** are the same as (**a**) except that NA/PA and NA/DCB were used, respectively, instead of NA/PCP. For delayed spectra in (**b**, **c**), $\Delta t = 50$ ms. **d** Time-resolved decay curves of NA/PCP, NA/PA, and NA/DCB at maximum phosphorescence emission ($\lambda_{ex} = 254$ nm) and at maximum fluorescence emission (insert, $\lambda_{ex} = 280$ nm) at ambient conditions. NA/PCP, NA/PA and NA/DCB were obtained using a melt-casting method with a starting mass ratio of NA: PCP = NA: PA = NA: DCB = 1:100, respectively. Melt-casting: heat up to the melting point ($T_m$) of the host to dissolve NA as colorless and clear solution, then cool to room temperature quickly. $T_{m(PCP)} = 126$ °C; $T_{m(PA)} = 131$ °C; $T_{m(DCB)} = 139$ °C. $\phi_{(F + P)}$: overall quantum yield; $\tau_F$: fluorescence lifetime; $\tau_P$: phosphorescence lifetime

overall quantum yield of NA/DCB was measured to be 23.0%. For all the guest/host system, the yellow URTP could be excited by a broad UV spectrum (Supplementary Figs. 16–18).

In addition to melt-casting, the ultralong yellow RTP of the guest/host system could also be obtained by grinding or evaporation methods. As shown in Supplementary Fig. 20, when grinding the host and guest together or co-dissolving the two component in ethyl acetate (EA) and then removing the solvent by evaporation, the obtained guest/host systems all show ultralong yellow RTP lasting for seconds, proving that melt-casting at relatively high temperature does not generate impurities that might be responsible for the new RTP band. Their steady-state, delayed PL spectra and lifetime decays (Supplementary Figs. 21 and 22) indicate that main fluorescence bands were centered around 400 nm and main phosphorescence bands were centered around 540 nm with ultralong lifetimes of hundreds of milliseconds. The variations in relative RTP/fluorescence intensities, lifetimes and slight wavelength shifts between different fabrication methods of the guest/host systems

might be due to phase separation and inhomogeneity at different extents.

To further verify the attribution of emission bands of the guest/host system, control experiments were carried out at low temperature and in different matrices. As shown in Supplementary Figs 6–12, PCP, PA, and DCB single crystals all exhibited strong blue phosphorescence at 77 K and their emission bands cover the range from 400 to 500 nm with highly resolved vibrational peaks. On the other hand, NA crystals exhibited yellow phosphorescence with a major emission peak at 558 nm ($\tau = 3.6$ ms) and a minor one at 617 nm together with vibrational progression. When NA/PCP, NA/PA, and NA/DCB were cooled to 77 K, besides the increased lifetimes around 540 nm (Supplementary Fig. 19), phosphorescence bands characteristic of the host (400–500 nm) and NA (558 nm) crystals, respectively, emerged in the PL spectra.

When NA was dispersed in polystyrene (PS), no RTP could be observed at all even under $N_2$ and only fluorescence emission at 395 nm ($\tau = 4.9$ ns) and 470 nm ($\tau = 6.3$ ns) was present in the

PL spectrum (Supplementary Fig. 14). However, when cooling to 77 K, persistent yellow phosphorescence could be noticed and new emission band at 536, 558, and 580 nm emerged (Supplementary Fig. 15). The 558 nm emission was ascribed to NA crystal while long delayed emission at 536 nm ($\tau = 606$ ms) and 580 nm were ascribed to NA dispersed in PS at the molecular level. On the other hand, the glassy NA solution in alcohol (ethanol/methanol, $v/v = 4/1$) at 77 K exhibited a fluorescence band with resolved vibrational modes at 354 nm, 367 nm ($\tau = 4.0$ ns), 388 and 410 nm, and a phosphorescence band with resolved vibrational modes at 496 nm, 534 nm ($\tau = 711$ ms), 578 and 629 nm (Supplementary Fig. 13). Based on these observations, it is rational that in the PL spectra of NA/PCP, NA/PA, and NA/DCB at room temperature, the set of peaks centered around 400 nm were ascribed to the vibration energy levels of the singlet excited state of NA dispersed at the molecular level (solid solution in hosts). On the other hand, the set of delayed emission peaks centered around 540 nm were ascribed to the vibration energy levels of the triplet excited state of NA dispersed at the molecular level, which accounts for the observed ultralong yellow RTP in the guest/host system (Fig. 1).

**ML properties of the guest/host system**. In addition to UV light, mechanical force could also serve as an excitation source for the yellow URTP in our guest/host system. As shown in Fig. 4 and Supplementary Movies 3–5, when crushing solid NA/PCP, NA/PA or NA/DCB obtained via melt-casting, a blue flashlight along with an ultralong yellow delayed emission lasting for seconds could be observed by naked eyes. The flash and delayed ML were strong enough to be noticed under soft room lighting (Fig. 4, inserted photos, left). To interpret the nature of the ML, their spectra were recorded at different times after mechanical force stimulation ceased. Spectra at $t_1$ for the guest/host solids were recorded immediately following the force stimulation. It is shown that, ML spectrum of NA/PCP at $t_1$ is almost identical to the steady-state PL spectrum of NA/PCP, exhibiting main emission peaks centered around 400, 418, 433, 480, 510, 540, 586, and 638 nm. The spectrum at $t_2$, shortly after the force ceased, only exhibits major peaks at 540, 586, and 638 nm, which is quite similar to the delayed PL spectrum. Therefore, the ultralong delayed yellow ML is ascribed to mechano-excited RTP of NA/PCP.

Similarly, ML spectra of NA/PA and NA/DCB also highly resemble their PL spectra, respectively, in terms of major emission peaks, and the set of peaks around 540, 586, and 638 nm at $t_2$ are also consistent with their mechano-excited yellow URTP. It is noted that after crushing the guest/host by repeated mechanical force, the initial large piece of solid turned into powder-like or much smaller pieces of solid and no ML could be observed upon further force stimulation. Their XRD data (Supplementary Fig. 24) indicate that they should become crystalline solids with much smaller sizes below which their ML vanish, which is similar to the single-component host counterparts[36,37]. Thanks to low melting point of the host molecules (all < 140 °C), the ML properties of the guest/host system could be restored upon facile melt-casting process and this ML off-on cycle could be reproduced for unrestricted times (Fig. 4d).

Mechano-excited organic URTP was rarely observed and the ultralong ML lasting for seconds to naked eyes in current guest/host system with excellent renewability may find applications in many fields concerning mechanical force sensing[43]. As for ML mechanisms, it is rational that the ML properties of the guest/host system resulted from their hosts since three hosts are all ML-active while two additional ML-inactive hosts (ABDO and TMBN) can only give rise to UV-excited URTP of NA (Supplementary Figs. 31 and 32). The change in ML spectra after NA dissolving are identical to the PL changes, comparing to the pure host crystals. Therefore, the newly emitting states are most likely to be identical in ML and PL after the guest/host structure was fabricated, despite different excitation sources.

**Ultrafast spectroscopy study**. Although the guest/host systems have been extensively examined in inorganic afterglow materials, the interpretation is far from sufficient in organic systems, where the guest RTP is usually considered to be a result of reduced non-radiative decay. It is noted in the experiment that NA does not always exhibit RTP in all matrices (such as PS) under deoxygenated conditions, we thus suspect that the host could also play an active role in the emergence of URTP observed in addition to diluting and rigidifying the guest. To shed light on the excited state dynamics concerning each component, fs-TA experiments were carried out.

As shown in Fig. 5, after excitation by the 267 nm pulsed laser, the resulted transient absorption spectra of pure PCP film exhibited a narrow band centered at 355 nm initially at 919 fs, which can be assigned to the lowest singlet excited state of PCP. This band decays over time and gives rise to a new broad peak at 636 ps with a maximum at 460 nm, assignable to the lowest triplet excited state (emitting state). As for NA/PCP film (Fig. 5d), an almost identical peak at 355 nm, initially at 955 fs, presumably the lowest singlet state of PCP, decays much more quickly than that of pure PCP. The duration to cross over to the lowest triplet excited state (the same broad peak centered at 460 nm) of PCP is a mere 126 ps. For comparison, the transient absorption spectra of pure PCP at 144 and 636 ps and that of NA/PCP at 126 ps were normalized in Fig. 5e. From the figure, the triplet band of NA/PCP at 126 ps already fully superimposes with that of the pure PCP at 636 ps except for the peak at ~370 nm caused by the presence of NA (Supplementary Fig. 26). The results clearly show that the ISC of PCP has been accelerated substantially upon dissolving trace amount of NA, which unambiguously shows that the excited-state dynamics for PCP is altered in the presence of NA. Usually, faster ISC is caused by enhanced spin-orbit coupling (SOC) and/or diminished energy gap between the two electronic states of different multiplicities. Although the carbonyl group has been known to help facilitate the ISC process, the minute amount of NA presence is not likely to cause any noticeable change to PCP since such external SOC enhancement requires orbital overlap between the guest and host molecules. Furthermore, the difference between the excitation spectrum of NA/PCP (monitored at 700 nm which belongs to the NA RTP judging from the vibrational pattern) and that of the pure PCP indicates that energy transfer from the host to the guest is not the major cause of the observed URTP (Supplementary Fig. 30). Experimentally, the tremendous effect could only mean that NA changes the photophysical dynamics of several PCP molecules in proximity simultaneously, given that detection by absorption requires new species of high concentrations. Our interpretation of the results is proposed in Fig. 5a, b, where a cluster of molecules consisting of PCP and NA are believed to have participated in the excited states. The cluster excitons thus serve as the experimentally detected transient species, which then rapidly decay to a lower and more stable triplet state, i.e., the localized triplet state of NA. The concept of cluster exciton has been reported for many naturally occurring events in nature, such as the photon harvesting process in photosynthesis, where multiple dye molecules are simultaneously excited to shorten the time required for energy transfer to the reaction center[44–46,50]. In organic guest/host systems however, such phenomenon is rarely explored.

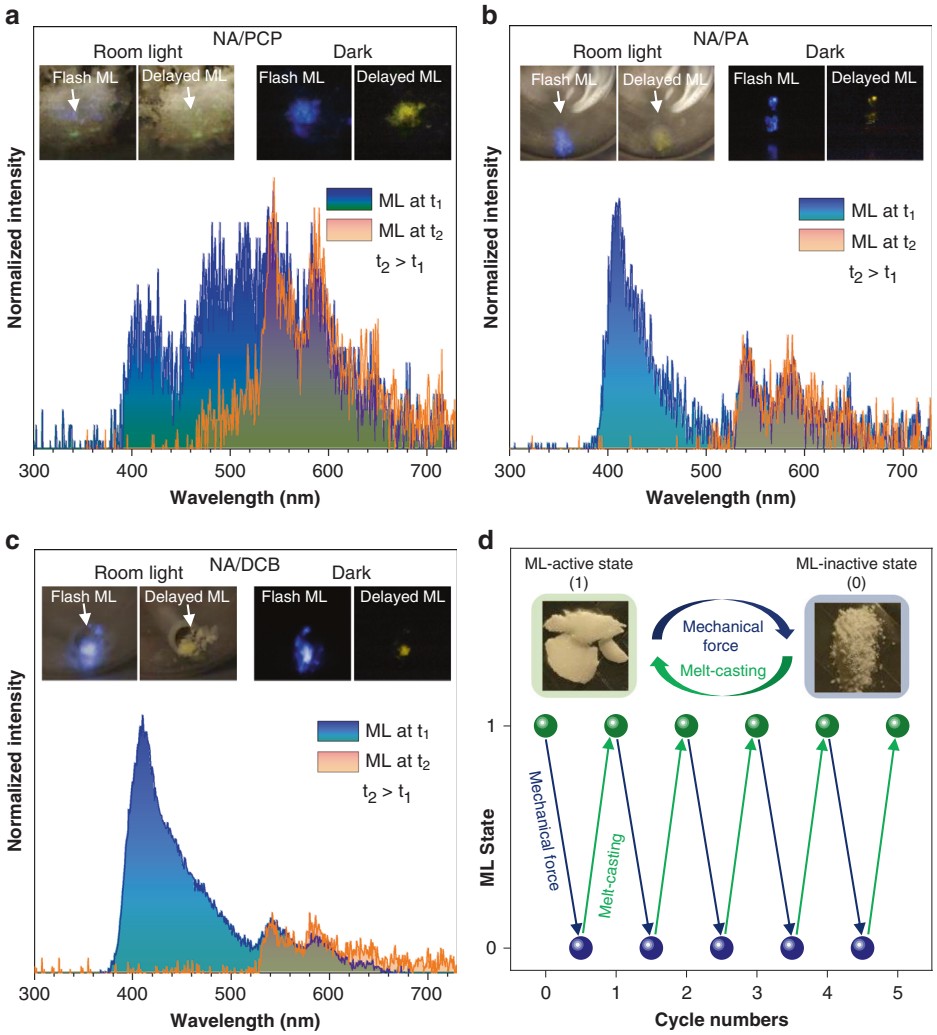

**Fig. 4** Mechanoluminescence (ML) properties of the guest/host system. **a** ML spectra of NA/PCP at different delay times ($t_1$ and $t_2$, $t_2 > t_1$). Insert (left): photos showing the flash-like ML of NA/PCP upon crushing the sample on the wall of a glass vial under soft room light and delayed ML after the crushing ceased. Insert (right): photos showing the flash-like and delayed ML of NA/PCP in the dark. **b**, **c** are the same as (**a**) except that NA/PA and NA/DCB were used, respectively, instead of NA/PCP. **d** Plot of ML state (1-on, 0-off) versus repeated mechanical force and melt-casting treating. Insert: photos showing the appearance of NA/PA after melt-casting (left) and crushing (right) and the repeated treating cycle

It has to be noted that the cluster exciton model is not only limited to an isolated example of NA/PCP. Although the ISC processes of PA and DCB could not be clearly detected by fs-TA measurements (Supplementary Figs. 28 and 29) possibly due to a dark triplet state, a similar ISC acceleration of the host in presence of NA was also observed for the NA/ABDO solid-state solution (Supplementary Fig. 33), where the state of NA/ABDO around 4 ps is almost identical to that of the pure ABDO at 17.6 ps.

If the proposed cluster exciton model were correct, two conditions should be met: (1) the singlet excited states of the guest and the host should be in resonance (energetically close) to form a cluster exciton; (2) the energy of the excited cluster should not be too far off either the guest (host) singlet excited state or guest triplet excited state, an essential prerequisite for efficient radiationless transitions[47]. The hypothesis was tested by three additional phosphors with varying singlet/triplet energy levels (PA, 2,3-NA, and NDA, Supplementary Figs. 34–36, Supplementary Table 1) in addition to NA using PCP as the host. Only 2,3-NA/PCP shows a green URTP with a duration of ~11 s, Supplementary Fig. 37), which matches both conditions. In contrast, both PA/PCP and NDA/PCP exhibit characteristic RTP of the PCP host (Supplementary Fig. 38).

As for the general guideline to choose the host in the cluster exciton model, it is reasoned that the host and guest should have a matching electron donating or withdrawing character to avoid the formation of a strong charge transfer state which tends to quench luminescence[48]. For instance, all effective hosts (PCP, PA, DCB, ABDO, and TMBN) that can turn on the URTP of NA contain electron-withdrawing group; in contrast, NA/DMB (dimethoxybenzene which is strongly electron-donating, Supplementary Fig. 31) did not show RTP at all. Finally, to suppress the competitive non-radiative decay of the cluster exciton, the host species should have a melting point sufficiently higher than room temperature.

**Applications**. Besides insights in mechanistic interpretation, we finally demonstrated the application values of current guest/host RTP system. Since the guest can be dissolved into the hosts at molecular level via facile evaporation and grinding methods, the system is of great potential to serve as security ink and scratch/ stress detection materials. Take NA/PCP for example (Fig. 5f, left), the part of Chinese character "He" (means summation, etc.) and the other part of a filter paper was soaked by EA solution of NA/PCP and pure PCP respectively and then dried and

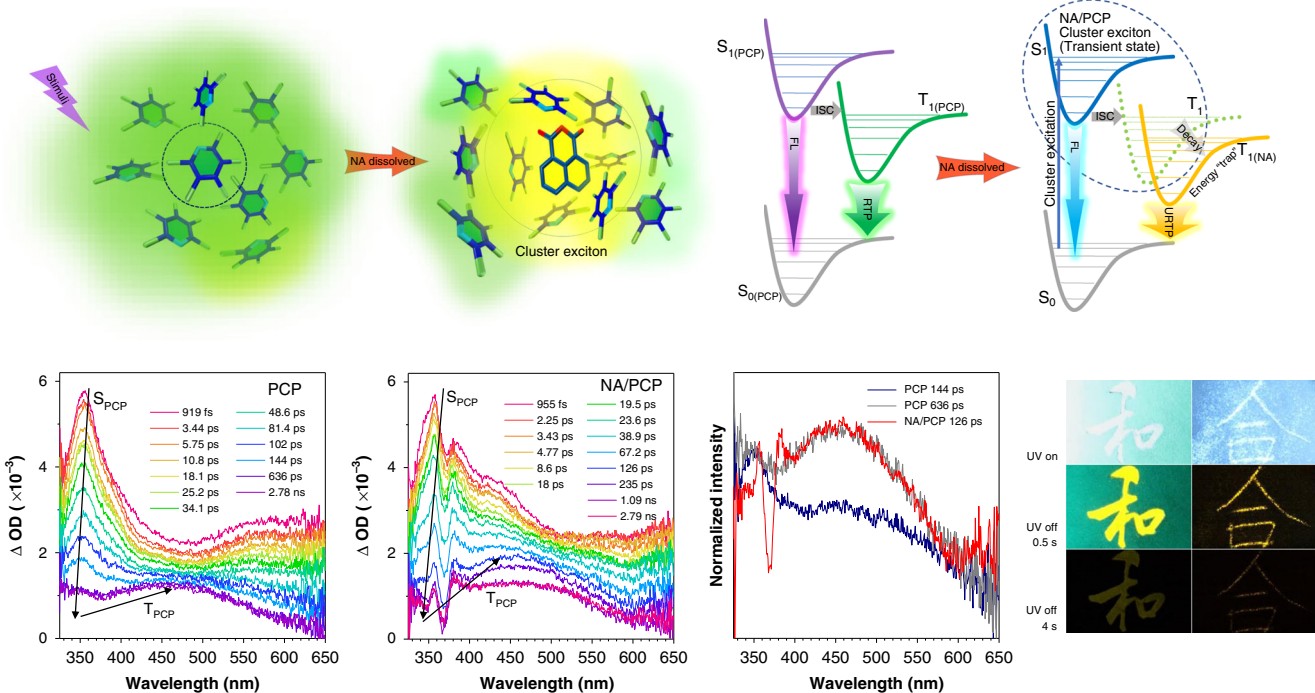

**Fig. 5** Mechanism and application of the guest/host system. **a** Left: schematic illustration of emission from pure PCP. Right: schematic illustration of emission from NA/PCP cluster exciton, where NA changes the photophysical dynamics of several PCP molecules in proximity. **b** Left: Jablonski diagram of the excitation and emission from pure PCP. Right: Jablonski diagram of the excitation and decay of the NA/PCP cluster exciton, i.e., a transient state consisted of NA and PCP molecules in proximity before the localized triplet state of NA acts as an energy trap for the URTP state. **c**, **d** Femtosecond transient absorption (fs-TA) spectra of PCP film and NA/PCP film, respectively, at different delay times after 267 nm excitation. **e** Normalized fs-TA spectra of NA/PCP film at a delay time of 126 ps, PCP film at delay time 144 ps and 636 ps. **f** Left: photos showing the potential of NA/PCP as a security ink material. The part of Chinese character "He" (means summation etc.) was soaked in EA solution of NA/PCP (wt/wt = 1/100) ($c_{PCP}$ = 10 g L$^{-1}$) and then dried quickly, the other part of the filter paper was soaked with EA solution of PCP (10 g L$^{-1}$) and then dried quickly. Right: photos showing the potential of NA/PA as a scratch detection material. NA/PA powder was obtained by gently mixing separately ground PA and NA powders. The powder was sprinkled evenly on a filter paper. Chinese character "He" (means combination, cooperation etc.) was written heavily using a cotton swab on the filter paper ($\lambda_{ex}$ = 254 nm)

combined at ambient conditions. The resulting paper looked like as a normal one without writing to naked eyes under ambient lighting. However, after 254 nm UV excitation, the part of "He" and the other part of the filter paper exhibited notable persistent yellow afterglow lasting for over 4 s and relatively shorter green afterglow lasting for only ~1 s, respectively. Therefore, dissolving in proper solvent, current guest/host system can serve as ideal security ink materials due to their distinguishable URTP. On the other hand, as shown in Fig. 5f (right), mixture of PA and NA powders was sprinkled evenly on the whole surface of a filter paper and then a Chinese character "He" (means combination, cooperation, etc.) was written by scratching heavily on the paper with a cotton swab. Under ambient lighting, the filter paper seemed normal without powder or scratch on it. After 254 nm UV excitation, however, the scratched/written part, i.e., the Chinese character, on the paper exhibited remarkable yellow afterglow lasting for over 4 s. Therefore, turn-on of the yellow URTP by simply dissolving the guest into the host under mechanical force (also seen in Supplementary Fig. 25) made them hopeful to serve as scratch or stress detection materials.

## Discussion

In this work, we have designed and achieved purely organic RTP with both high efficiency and ultralong lifetime utilizing a guest/host strategy. When the host was ML-active, the URTP of the guest/host system could also be produced by mechanical excitation. Furthermore, we succeeded in elucidating the RTP photophysics in solid state by means of ultrafast spectroscopy (fs-TA). The fs-TA studies revealed that: (1) host and guest molecules worked synergistically in the photo-excited electronic transition processes; (2) cluster excitons spanning both guest and host molecules formed as transient species before guest molecules act as an energy trap, localized triplet excited state, to emit RTP. Therefore, besides the passive role of the host in suppressing non-radiative decay of the guest, we showed that host and guest species both played active roles in the photophysics. In this sense, mechanistically similar RTP, vs. inorganic afterglow, exists in purely organic systems. The difference is that the afterglow of inorganic materials arises from a charge separated state while the ultralong luminescence in our system is ascribed to emission from a localized triplet excited state, i.e., molecular phosphorescence. Combined advantages of current purely organic guest/host URTP system, such as simplicity, low cost, scalability, absence of unstable species (C–Br and C–I bond), multiple excitation sources, facile fabrication and excellent performances, are expected to show great values in practical applications. In addition, mechanistic insights in this work may advance the understanding of organic phosphorescence and help develop more URTP materials in the future, basing on unlimited combinations of host and guest species.

## Methods

**Materials**. Chemicals were purchased from Sigma-Aldrich and were purified by column chromatography followed by triple recrystallization, and their purity were verified by HPLC (Agilent 1260 Infinite HPLC system with ZORBAX SB-C18 column) measurement before optical characterizations. The flow rate was fixed at 1.0 mL min$^{-1}$, the injection volume was 20 μL. The absorption wavelength used was set at 280 nm. 100% percent of acetonitrile was used as the running buffer. Solvents were obtained from Sigma-Aldrich and were dried by distillation before used.

**Photoluminescence (PL) measurements.** The steady-state, delayed PL spectra and phosphorescence lifetime were measured on an Edinburgh FLSP 980 fluorescence spectrophotometer equipped with a xenon arc lamp (Xe900) and a microsecond flash-lamp (uF900). Absolute PL quantum yields were measured using a Hamamatsu absolute PL quantum yield spectrometer C11347 Quantaurus_QY. Delayed emission of NA/PCP ($\Delta t = 1$ s) was recorded on an Ocean Optics Spectrometer (QE Pro).

**Mechanoluminescence (ML) measurements.** The ML spectra were collected from a spectrometer of Acton SP2750 with a liquid-nitrogen-cooled CCD (SPEC-10, Princeton) as a power detector, which were further checked by an Ocean Optics Spectrometer (QE Pro).

**Photos and videos.** The photos and videos were recorded by a Cannon EOS 60D camera.

**Femtosecond transient absorption (fs-TA) experiments.** The fs-TA measurements were accomplished using a femtosecond regenerative amplified Ti:sapphire laser system in which the amplifier was seeded with the 120 fs laser pulses from an oscillator laser system[49]. The laser probe pulse was produced by utilizing ~5% of the amplified 800 nm laser pulses to generate a white-light continuum (325–650 nm) in a $CaF_2$ crystal, and then this probe beam was split into two parts before traversing the sample. The film was excited by a 267 nm pump laser beam. The samples for fs-TA experiments were prepared by melting 1 mg sample placed between two quartz slides (3 cm × 3 cm) and then cooling to room temperature quickly.

## Data availability

The authors declare that all data supporting the findings of this study are available within this article and Supplementary Information files, and also are available from the authors upon reasonable request.

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

## Acknowledgements

We are grateful for financial support from the National Science Foundation of China (51622305, 21788102, and 51873092), the Research Grants Council of Hong Kong (16308116, C6009-17G, and A-HKUST605/16), the Innovation and Technology Commission (ITC-CNERC14SC01 and ITCPD/17-9), the open fund of State Key Laboratory of Luminescent Materials and Devices in South China University of Technology (2018-skllmd-05), the National Key R&D Program of China (2017YFA0303500), the Fundamental Research Funds for the Central Universities (WK2340000068), the Hong Kong Research Grants Council (GRF 17302218) and The University of Hong Kong Development Fund 2013-2014 project "New Ultrafast Spectroscopy Experiments for Shared Facilities". X.Z. thanks Mr. Junyi Gong, Hao Xing, Peifa Wei, Haoke Zhang, Jianguo Wang, Zheng Zheng, Pengfei Zhang, Shunjie Liu, Haitao Feng, Yong Liu, Benzhao He, Nelson L.C. Leung, Jie Yang, Xinyuan He, Wenjie Wu, Zhiyao Hou, Han Zhang and Miss Huiqing Peng, Qian Xie, Yujie Tu, Jing Zhang, Fan Liao, and Huijun Liu for their kind help and discussions.

## Author contributions

X.Z. designed the RTP system, finished the data acquisition, analysis, and paper writing; L.D. and D.L.P. assisted in fs-TA measurements and the data analysis; W.Z. assisted in photophysical property measurements and provided helpful discussions. Thus, X.Z., L.D., and W.Z. contributed equally to this work. Z.Z. and Y.X. assisted in photophysical property measurements and provided helpful discussions. X.H. assisted in HPLC measurement and discussion. P.F.G. assisted in photo and movie taking. P.A. provided helpful discussions. C.W. and Z.L. assisted in mechanoluminescence measurements. J.L., J.X.L., and C.Z. provided helpful discussions on fs-TA measurements. J.W.Y.L. revised the paper. G.Z. assisted in mechanoluminescence measurements, supervising fs-TA measurement, proposing the cluster exciton model and revising the paper. B.Z.T. is responsible for the funding acquisition and supervised this work.

## Competing interests

The authors declare no competing interests.
