## [Peer Review File · Nature Communications]

Reviewers' comments:

Reviewer #1 (Remarks to the Author):

The Manuscript is really interesting and inspiring and it definitely adds some of the missing pieces to the intricate puzzle of organic RTUP.

The work is worth of publication on Nature Communication after better clarifying some details:

"The solid-state solution of NA in PCP was fabricated via melt-casting. A blend of NA and PCP (wtNA:wtPCP = 1:100) was heated just over the melting point of PCP (124 °C), where a colorless, clear solution of NA in liquid PCP formed and was cooled to room temperature quickly. The resulting white solid under room lighting was denoted as NA/PCP."

How is the blend prepared? Mixing powders of the two components manually? In a ball miller? How is the clear solution quickly cooled? How are these systems different from films prepared for fs-TA experiments? Have the different NA/host solids been characterized by XRD?

"It is noted that after crushing the guest/host by repeated mechanical force, the initial large piece of solid turned into powder-like solid and no ML could be observed upon further force stimulation." Is there a XRD characterization before and after the grinding? I have found XRD patterns in the SI but, since they are not mentioned in text, their interpretation is difficult. Is PCP ML only in the form of single crystals or also as powders? This information is important to relate the ML properties of the host and NA/host solids

Reviewer #2 (Remarks to the Author):

A new "cluster exciton" concept was proposed in organic guest/host system exhibiting marvelous URTP and mechanoluminescence with long lifetime. The simple mixture of trace amount of NA with three host molecules through melt-casting, grinding or evaporation methods could bring out intriguing mechanoluminescence and photoluminescence properties. Pretty detailed data and convincing discussion were made, and the femtosecond transient absorption experiments were also utilized to elucidate the RTP photophysics in solid state for the first time. I think this manuscript could be accepted in Nature Communications after addressing the following minor points:

1. Except for the "cluster exciton" to enhance the ISC process, I wonder if there is any possibility for energy transfer in this host/guest system due to the overlap of the absorption spectra of the guest and the photoluminescence spectra of the host. The author should clarify and prove this point.
2. In fig 2. The PL spectra was examined at $\lambda_{ex} = 254$ nm, but the decay curves was examined at $\lambda_{ex} = 280$ nm. Please clarify this point.
3. Indeed, the presence of NA accelerated the ISC of PCP. Does NA affect the ISC process of PA and DCB in the same way?
4. What are the requirements or standards to choose host species?
5. Have the authors tried using other phosphors except NA? Are there any matching conditions between host and guest species?
6. As shown in Fig. 2a and Fig. S6a, the delayed PL spectra of PCP crystal at room temperature and 77 K are different. Why?
7. Are the guest/host systems still in crystalline state after melt-casting? Relevant characterizations should be performed.
8. Some typing errors are listed below:
 - a. Page 6, 1st line, "Fig 2d-right" should be "Fig 2b-right".
 - b. Page 9, 5th line, "510 nm".
 - c. Page 14, 5th line from the bottom, "636 ps".
9. More related recent RTP references are recommended to cite: J. Am. Chem. Soc., 2018, 140, 1916; Acc. Chem. Res., 2019, 52, 738.

Reply to Reviewers' Comments for Manuscript NCOMMS-19-14750

Responses to the Comments and Suggestions of Reviewer 1

General comment: *The manuscript is really interesting and inspiring and it definitely adds some of the missing pieces to the intricate puzzle of organic URTP. The work is worth of publication on Nature Communication after better clarifying some details.*

Our reply: We greatly appreciate the positive comments from the reviewer on our work. Indeed, as the reviewer pointed out, this work will add some of the missing pieces to the intricate puzzle of organic URTP. Our point-by-point response is given below.

Comment 1: *“The solid-state solution of NA in PCP was fabricated via melt-casting. A blend of NA and PCP (wt_{NA}:wt_{PCP} = 1:100) was heated just over the melting point of PCP (124 °C), where a colorless, clear solution of NA in liquid PCP formed and was cooled to room temperature quickly. The resulting white solid under room lighting was denoted as NA/PCP.” How is the blend prepared? Mixing powders of the two components manually? In a ball miller? How is the clear solution quickly cooled? How are these systems different from films prepared for fs-TA experiments?*

Our reply: We thank the reviewer for suggesting that we should provide more important details about the melt-casting process. We have now added more detailed description along with figures in the revised supplementary information as follows:

“The initial states of the host and guest molecules are both crystalline after they were purified by silica gel column chromatography and subsequent recrystallization (3×). Take NA/PCP for example, to make the solid-state solution as homogeneous as possible, before melt-casting, the crystals of NA and PCP (wt_{NA}:wt_{PCP} = 1:100) were first ground together in an agate mortar into a fine powdery mixture. As shown in the following photos, the powder was then transferred onto a quartz slide which was placed on a hot plate stirrer for heating (T = 130 °C). Upon the melting of the host (T_m = 126°C), a colorless and clear solution to the naked eye formed (at a relatively high temperature, the liquid/melted host should have served as a good solvent to dissolve the significantly less amount of NA). Then the hot quartz slide was placed on a test bench at room temperature. Once the melt was cooled to a temperature lower than the T_m of PCP, a white solid quickly formed. The larger pieces of the white solid (not transparent to naked eyes) were then used for PL and ML measurements.”

“The samples for fs-TA measurements were also prepared via melt-casting similar to the above description. The difference is that the sample for fs-TA measurements

was made into largely transparent thin films, since fs-TA measurements require decent transparency. Specifically, a tiny amount (~1 mg) of NA/PCP powder sample was placed in between two quartz slides (3 cm × 3 cm) which was then placed on a hot plate stirrer (T = 130 °C) to melt. Shortly after the hot quartz slides were placed on the test bench at room temperature, the liquid sample quickly became transparent film-like solid in between the two quartz slides, which were later used for fs-TA measurements.”

Comment 2: *Have the different NA/host solids been characterized by XRD? “It is noted that after crushing the guest/host by repeated mechanical force, the initial large piece of solid turned into powder-like solid and no ML could be observed upon further force stimulation.” Is there an XRD characterization before and after the grinding? I have found XRD patterns in the SI but, since they are not mentioned in text, their interpretation is difficult. Is PCP ML only in the form of single crystals or also as powders? This information is important to relate the ML properties of the host and NA/host solids.*

Our reply: We thank the reviewer’s important questions about the relationship between ML properties and the state of crystals. We have provided XRD characterizations for the NA/host solids in the supplementary information. Indeed, what we observed is that when the host crystals (PCP, PA or DCB) or the solid-state solutions prepared from melt-casting are crushed by repeated mechanical force, they turned into much smaller pieces of solids or even powders. When they are large solid pieces, ML is very easy to observe under crushing or grinding forces. While the ML could hardly be observed when the sample turned into much smaller pieces or powders. As shown in Supplementary Figures 23 and 24, in most cases, the powders obtained from repeated crushing or grinding show additional peaks than their intact counterparts in terms of the XRD patterns, indicating that they are not amorphous. Instead, they should also be crystals with much smaller sizes and more complicate surface structures. It is known that the signal of XRD experiments depends heavily on surfaces of the samples. Powder-like crystals with very small sizes will provide larger specific surface area and thus show more peaks in the XRD data. In short, the XRD data in our cases showed that the ML-inactive samples were small crystals (powder-like to the naked eye) rather than amorphous solids. Our results are consistent with previous studies concerning ML dependence on crystal sizes based on observations of numerous ML molecules: “...decrease in intensity as the crystal size is decreased...” (*J. Am. Chem. Soc.* 1973, 95, 7510-7512) and “For every substance, there exists a crystal size below which the substance is not mechanoluminescent.” (*Luminescence* 2014, 29, 977-993).

As for PCP, it only showed visible green ML when the large crystals were crushed. As fine powders, their ML was hard to observe. In summary, similar to the NA/host solids, the ML property of the PCP crystal is also size-dependent.

According to the suggestions of the reviewer and based on the above analysis, we have added a discussion of XRD data and the relationship between ML property and crystal sizes, and relevant literature (refs 36 and 37) in the revised manuscript. It reads “When turned into fine powders after repeated crushing, PCP hardly exhibits ML after being subjected to further mechanical forces. The XRD data of the PCP powder (Supplementary Figure 23) show extra peaks than do those of the PCP single crystals, indicating that the PCP powders are crystals with much smaller sizes rather than amorphous. Therefore, it is concluded below a certain limit of crystal size, the PCP solid is no longer mechanoluminescent.^{36,37}” (Line 163) “Their XRD data (Supplementary Figure 24) indicate that they should become crystalline solids with much smaller sizes below which their ML vanish, which is similar to the single-component host counterparts.^{36,37}” (Line 371)

Responses to the Comments and Suggestions of Reviewer 2

General comment: *A new “cluster exciton” concept was proposed in organic guest/host system exhibiting marvelous URTP and mechanoluminescence with long lifetime. The simple mixture of trace amount of NA with three host molecules through melt-casting, grinding or evaporation methods could bring out intriguing mechanoluminescence and photoluminescence properties. Pretty detailed data and convincing discussion were made, and the femtosecond transient absorption experiments were also utilized to elucidate the RTP photophysics in solid state for the first time. I think this manuscript could be accepted in Nature Communications after addressing the following minor points.*

Our reply: *We are grateful to the reviewer of the positive comments on our work. Accordingly, we revised the manuscript point by point as follows.*

Comment 1: *Except for the “cluster exciton” to enhance the ISC process, I wonder if there is any possibility for energy transfer in this host/guest system due to the overlap of the absorption spectra of the guest and the photoluminescence spectra of the host. The author should clarify and prove this point.*

Our reply: We thank the reviewer very much for this valuable question. Accordingly, by comparing the excitation spectra of corresponding species, we have strong evidence that most of the observable URTP of the NA/PCP solid state solution did not result from energy transfer from the host (PCP). From the delayed emission spectrum, it is clear that NA/PCP shows a vibrational progression around 700 nm while pure PCP crystal exhibits only signal baseline at this particular wavelength. As a result, the excitation spectrum of NA/PCP was recorded when monitored at $\lambda_{em} = 700$ nm, which should give the origin of the NA/PCP URTP. Meanwhile, the excitation spectrum of PCP crystal at the RTP maximum ($\lambda_{em} = 510$ nm) has been provided in Supplementary Figure 5. As shown in Supplementary Figure 30, the excitation spectra of NA/PCP at

$\lambda_{em} = 700$ nm and PCP at $\lambda_{em} = 510$ nm are very dissimilar. This result should strongly support against the hypothesis that the URTP of NA/PCP solid state solution is a result of energy transfer from the excited-state PCP molecules.

Supplementary Figure 30. Excitation spectra of the NA/PCP solid monitored at $\lambda_{em} = 700$ nm and the pure PCP solid monitored at $\lambda_{em} = 510$ nm at room temperature, respectively.

We have added related discussion in the revised manuscript and it reads “**The difference between the excitation spectrum of NA/PCP (monitored at 700 nm which belongs to the NA RTP judging from the vibrational pattern) and that of the pure PCP indicates that energy transfer from the host to the guest is not the major cause of the observed URTP (Supplementary Figure 30)**” (Line 443)

Comment 2: *In fig 2. The PL spectra was examined at $\lambda_{ex} = 254$ nm, but the decay curves were examined at $\lambda_{ex} = 280$ nm. Please clarify this point.*

Our reply: We thank the reviewer for raising this point. The RTP properties of the samples were firstly observed by the naked eye when they are excited under a 254-nm handheld UV lamp. Therefore, to make the measurement condition as close as visual observations, we selected $\lambda_{ex} = 254$ nm as the excitation wavelength. However, such wavelength is not available in fluorescence lifetime measurement which utilizes LED lasers. We thus recorded the fluorescence lifetime decay curves using an excitation wavelength closest to 254 nm, i.e., $\lambda_{ex} = 280$ nm.

Comment 3: *Indeed, the presence of NA accelerated the ISC of PCP. Does NA affect the ISC process of PA and DCB in the same way?*

Our reply: We thank the reviewer for raising this important question. As provided in the supplementary information (Supplementary Figures 28 and 29), the ISC process of PA or DCB was not fully detected possibly due to a dark triplet state. However, to demonstrate the generality of the “cluster exciton” model, we performed additional

fs-TA measurements of another guest/host combination, i.e., NA/ABDO, and the pure ABDO solid (Supplementary Figures 31 and 33). It is shown that upon addition of a trace amount of NA, the ISC of ABDO is accelerated compared to that of the pure ABDO. Therefore, we believe that the “cluster exciton” mechanism is general rather than an isolated example.

Supplementary Figure 33. Femtosecond transient absorption spectra of ABDO film (a) and NA/ABDO film (b) respectively at different delay times. Excitation: 267 nm laser pulse.

The discussions of fs-data of NA/PA, NA/DCB and NA/ABDO are now provided in the revised manuscript. It reads “It has to be noted that the ‘cluster exciton’ model is not only limited to an isolated example of NA/PCP. Although the ISC processes of the PA and DCB could not be clearly detected by fs-TA measurements (Supplementary Figure 28,29) possibly due to a dark triplet state, a similar ISC acceleration of the host in presence of NA was also observed for the NA/ABDO solid state solution (Supplementary Figure 33), where the state of NA/ABDO around 4 ps is almost identical to that of the pure ABDO at 17.6 ps.” (Line 458)

Comment 4: *What are the requirements or standards to choose host species?*

Our reply: We thank the reviewer for giving us the opportunity to provide more detailed requirements for the host species. We have added relevant data concerning two more hosts than can “turn on” the RTP of NA: 6-Acetyl-1,4-benzodioxane (ABDO) and 3,4,5-Trimethoxybenzotrile (TMBN). As shown in Supplementary Figure 32, NA/ABDO and NA/TMBN solid-state solutions both show yellow RTP with a duration of ~6 s and a lifetime of hundreds of milliseconds. Their delayed emission spectra were similar to those of NA/PCP, NA/PA, and NA/DCB, showing characteristic vibrational peaks of the triplet state of NA dissolved in the host at the molecular level. We also provide an example where the host quenches the NA RTP, which is when the host was changed to dimethoxybenzene (DMB), no afterglow

could be observed at all.

We also noted that the guest molecule (NA) is very electron-deficient, and all hosts that turn on RTP also contain electron-withdrawing moieties (e.g., pyridine ring, carbonyl or cyano group). In contrast, DMB is an electron-rich molecule. Therefore, charge or electron transfer between DMB and NA may occur and thus quenches luminescence.

Combining the abovementioned results with those in the main text, the standards to choose host species to produce URTP are concluded as follows.

- (1) Electron-withdrawing or electron-donating characters of the host and guest species should match with each other to avoid the formation of a quenching intermolecular CT state. The phenomenon is well known due to large nuclear reorganization in the excited state.
- (2) The melting point of the host should be sufficiently higher than room temperature so that the competitive thermal quenching pathway is prohibited to allow RTP to be observed.

Accordingly, we have added relevant description concerning the standards of the host in the revised manuscript. It reads “As for the general guideline to choose the host in the “cluster exciton” model, it is reasoned that the host and guest should have a matching electron donating or withdrawing character to avoid the formation of a strong charge transfer state which tends to quench luminescence.⁴⁹ For instance, all effective hosts (PCP, PA, DCB, ABDO and TMBN) that can “turn on” the URTP of NA contain electron-withdrawing group; in contrast, NA/DMB (dimethoxybenzene which is strongly electron-donating, Supplementary Figure 31) did not show RTP at all. Finally, to suppress the competitive non-radiative decay of the ‘cluster exciton’, the host species should have a melting point sufficiently higher than room temperature.” (Line 474)

Comment 5: *Have the authors tried using other phosphors except NA? Are there any matching conditions between host and guest species?*

Our reply: We thank the reviewer for raising this important question. As illustrated in Fig 5b, the mechanism to induce the URTP of current guest/host system is based on the proposed “cluster exciton” model. The guest accelerates the ISC of several host molecules in proximity and the “cluster exciton” then serves as a transient state which rapidly decays to a lower and more stable triplet state, i.e., the localized triplet state of guest. As a result, two important factors should be considered: 1) the singlet excited states of the guest and the host should be in resonance (close by) to form the “cluster exciton”; 2) the “cluster exciton” should not be too far off the singlet excited states and guest triplet state (somewhere in between but not too far off either, an essential prerequisite for radiationless transitions, ref48).

For example, four phosphors (PA, 2,3-NA, NA and NDA, Supplementary Figure 34) with different triplet-state energy levels were compared when PCP was used as the

host. Similar to NA (Supplementary Figure 13), the singlet-state and the triplet-state energy levels of the PA, 2,3-NA and NDA were determined in rigid glass matrix (MeOH/EtOH, v/v, 4/1) at 77 K. The phosphorescence colors of PA, 2,3-NA, NA and NDA in alcohol solutions at 77 K were blue, green, yellow and orange, respectively (inserted photos in Supplementary Figures 9, 13, 35 and 36). Correspondingly, obtained from the steady-state and delayed emission spectra, PA, 2,3-NA, NA and NDA in alcohol solutions at 77 K show maximal fluorescence peaks at 350 nm, 377 nm, 367 nm and 379 nm and phosphorescence emissions at 439 nm, 511 nm, 534 nm and 560 nm, respectively. As for the PCP crystals, the fluorescence and phosphorescence are at 350 nm and 510 nm, respectively (Fig 2a). Apparently, PA is not a good guest since its triplet state is higher than that of the host, and it is impossible to tell which portion of the observed RTP is due to the “cluster exciton” or the host itself. In the case of 2,3-NA and NA, the singlet excited states are not too far off and so are the triplet ones. For NDA, however, its triplet state is perhaps too low to allow for efficient radiationless transitions from the cluster to the guest excited state.

In summary, we added in the revised manuscript that: “If the proposed ‘cluster exciton’ model were correct, two conditions should be met: 1) the singlet excited states of the guest and the host should be in resonance (energetically close) to form a ‘cluster exciton’; 2) the energy of the excited cluster should not be too far off either the guest/host singlet excited state and guest triplet excited state, an essential prerequisite for efficient radiationless transitions.⁴⁸ The hypothesis was tested by three additional phosphors with varying singlet/triplet energy levels (PA, 2,3-NA and NDA, Supplementary Figures 34-36, Supplementary Table 1) in addition to NA using PCP as the host. Only 2,3-NA/PCP shows a green URTP with a duration of ~11 s, Supplementary Figure 37), which matches both conditions. In contrast, both PA/PCP and NDA/PCP exhibit characteristic RTP of the PCP host (Supplementary Figure 38).” (Line 464)

Comment 6: *As shown in Fig. 2a and Fig. S6a, the delayed PL spectra of PCP crystal at room temperature and 77 K are different. Why?*

Our reply: We thank the reviewer for providing us the opportunity to give more detailed explanation on PCP’s temperature-dependent phosphorescence. As shown in Fig 2a and Supplementary Figure 6, the RTP and low-temperature (77 K) phosphorescence emissions of the PCP crystal were green and blue, respectively. Consistently, the main RTP peaks of PCP are located at 480 nm and 510 nm; instead, the maximal phosphorescence peak was around 440 nm ($\tau = 26$ ms) while peaks at 480 nm and 510 nm became shoulders in the emission spectrum at 77 K. We attribute this blue-shifted phosphorescence to the restriction of vibrational modes/thermal relaxations of PCP molecules at 77 K. It is common that at such low temperature as 77 K, molecular motions could be highly restricted which will reduce the possibility of internal conversion and thus enhance the possibility for radiative decay from the higher excited states to the ground state. This is also known as the “rigidochromic

effect". Indeed, the suppression of the thermal relaxation of triplet excited states of PA and DCB crystals are also obvious at 77 K, as indicated by their intense blue phosphorescence at this temperature (Supplementary Figure 8 and Supplementary Figure 10).

We have added relevant interpretation in the revised manuscript and it reads “At 77 K (Supplementary Figure 6), PCP crystals exhibited blue-shifted phosphorescence (Max $\lambda_{em} = 440$ nm) which is likely due to reduced thermal relaxation from higher triplet excited states. (Line 153)

Comment 7: *Are the guest/host systems still in crystalline state after melt-casting? Relevant characterizations should be performed.*

Our reply: We thank the reviewer’s valuable question which is similar to that of the referee 1. We have provided powder XRD data in the supplementary information. As shown in Supplementary Figure 24, all the guest/host systems still show crystalline patterns after melt-casting and thus they are still crystalline instead of amorphous.

Comment 8: *Some typing errors are listed below:*

a. Page 6, 1st line, “Fig 2d-right” should be “Fig 2b-right”.

b. Page 9, 5th line, “510 nm”.

c. Page 14, 5th line from the bottom, “636 ps”.

Our reply: The careful review from the referee is highly appreciated. We have made corrections accordingly and addressed relevant typing errors.

Comment 9: *More related recent RTP references are recommended to cite: J. Am. Chem. Soc., 2018, 140, 1916; Acc. Chem. Res., 2019, 52, 738.*

Our reply: We thank the referee very much for suggesting the citation of these representative publications in the field of organic RTP. We have added relevant literature as ref13 and ref15 respectively in the revised manuscript.

REVIEWERS' COMMENTS:

Reviewer #1 (Remarks to the Author):

The points raised by both Reviewers have been fully addressed by the authors. The Manuscript certainly deserves publication on Nature Communications

Reviewer #2 (Remarks to the Author):

The authors considered our previous comments and have revised the manu accordingly with detailed data and discussion. This manu can be now accepted for publication.

REVIEWERS' COMMENTS:

Reviewer #1 (Remarks to the Author):

The points raised by both Reviewers have been fully addressed by the authors. The Manuscript certainly deserves publication on Nature Communications

Response: We thank the reviewer for recognition of our revised manuscript.

Reviewer #2 (Remarks to the Author):

The authors considered our previous comments and have revised the manu accordingly with detailed data and discussion. This manu can be now accepted for publication.

Response: We thank the reviewer for recognition of our revised manuscript.